# *X*-Planes: Adaptive and Efficient Representation for Dynamic Reconstruction and Rendering in the Age of Large Pretrained Models

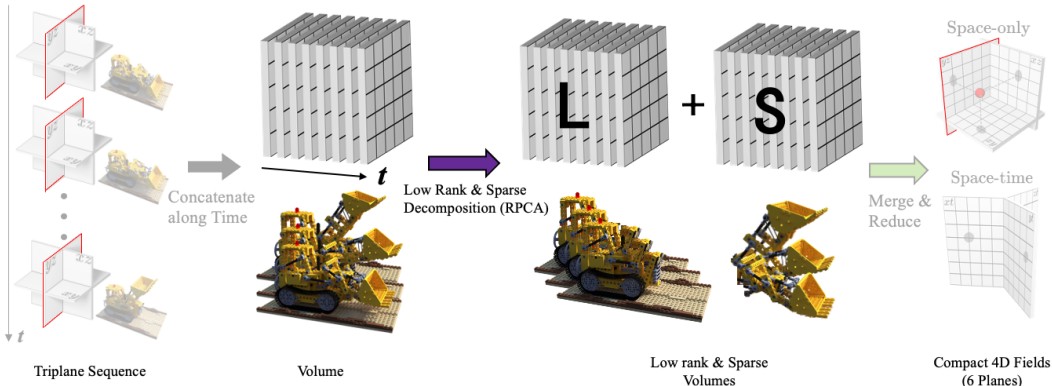

Figure 1: **Illustration of the method**. We employ the Large Reconstruction Model (LRM) to obtain an initial 4D representation, and decompose it into compact feature planes by low-rank and sparse decomposition (Robust PCA). The decomposed results serve as an effective initialization, improving performance for plane-based methods, such as *K*-Planes (Fridovich-Keil et al., 2023) and Tensor4D (Shao et al., 2023).

## ABSTRACT

In this paper, we present a novel dynamic NeRF pipeline with an effective initialization and distillation/optimization strategy. Previous approaches, such as *K*-Plane and Tensor4D, rely on randomly initialized compact feature plane representations to model 4D dynamic scenes, grounded in tensor decomposition theory. In contrast, our method employs a pre-trained Large Reconstruction Model (LRM) to generate a noisy and incomplete initial 4D representation, subsequently factorizes it into compact feature planes via low-rank and sparse decomposition, and reuses the feature decoder of LRM to initialize the NeRF MLP. The decomposed feature planes and decoder serve as both an effective initialization and regularization for the dynamic NeRF optimization, enabling state-of-the-art results with enhanced performance. The pipeline is broadly applicable to dynamic NeRF methods, and readily benefits from future advancements of the LRM, paving the way for more generalizable dynamic NeRF tasks.

## 1 INTRODUCTION

High-quality and fast reconstruction & rendering of dynamic scenes from a time series of captured images are crucial for applications such as AR/VR (Kanade et al., 1997), 3D content production (Starck and Hilton, 2007), and immersive entertainment (Collet et al., 2015). Traditional approaches rely on classical mesh-based representation for dynamic scene reconstruction. However, these methods are prone to reconstruction errors and rendering artifacts, especially when dealing with thin structures, specular surfaces, or topological changes (Shao et al., 2023).

Neural radiance fields (NeRF) has seen significant advancements since its introduction as a 3D representation (Mildenhall et al., 2021). Considerable efforts have been made to extend NeRF to dynamic scenes, which can be categorized into two main research lines: 1) incorporating time as an additional input dimension to model a 4D NeRF representation (Fridovich-Keil et al., 2023; Shao et al., 2023; Park et al., 2021a;b), and 2) disentangling a dynamic scene into a canonical radiance field and a dynamic motion field (Pumarola et al., 2021; Liu et al., 2022; Fang et al., 2022a; Gan et al., 2024). In NeRF-based methods, the current state-of-the-art (SOTA), such as $K$-Planes (Fridovich-Keil et al., 2023) and Tensor4D (Shao et al., 2023), represent the 4D field using 6 or 9 planes, enabling both a compact representation and efficient reconstruction of dynamic scenes. With recent developments in large-scale pretraining for 3D generation (Hong et al., 2024; Zou et al., 2024; Tochilkin et al., 2024), a natural question arises:

*What if we could start with an incomplete 4D field, rather than random initialization?*

The recently proposed Large Reconstruction Model (LRM) (Hong et al., 2024) takes an image as input and predicts its 3D representation in the form of a triplane, presenting new opportunities for triplane-based NeRF methods. For dynamic NeRF, LRM allows us to generate the triplanes for each time frame from monocular video in just a few seconds. These triplanes can then be stacked up to form an initial 4D field. However, this 4D field, especially in regions not captured by the monocular video, may be noisy and incomplete. In this paper, we focus on how to derive an initial compact representation, such as the 6/9 planes used in $K$-Planes (Fridovich-Keil et al., 2023) and Tensor4D (Shao et al., 2023), by decomposing this noisy and incomplete 4D field. Besides that, the pre-trained feature decoder (an MLP) in the LRM serves as the renderer (MLP) of the subsequent dynamic NeRF, further optimized on the dynamic scene dataset. The decomposed fields with the pre-trained renderer provide both a fast initialization and an effective regularization for subsequent dynamic NeRF optimization, improving the SOTA dynamic NeRF methods (Fridovich-Keil et al., 2023; Shao et al., 2023).

The existing NeRF-based tensor decomposition methods (Shao et al., 2023; Chen et al., 2022; Gao et al., 2023) primarily provide a theoretical foundation for compact representations (e.g. vector & matrix (Chen et al., 2022), 3 space planes (Chan et al., 2022), and 6 or 9 time-incorporated planes (Fridovich-Keil et al., 2023; Shao et al., 2023)). These methods initialize compact representations randomly and then directly optimize them in a data-driven manner, so the final features are not directly tied to the initial fields. In contrast, our approach leverages a decomposition that compresses the representation while preserving the quality of the initial 4D field. To achieve this, we introduce Robust Principal Component Analysis (RPCA) for 4D field decomposition. As illustrated in the lego bulldozer example (Figure 1), the static body and floor are low-rank components, while the moving shovel is represented as sparse components. We reduce the low-rank results to 3 spatial planes, which serve as a canonical space representation. For the sparse components, we treat them as the supervision for guiding the training of space-time planes during the subsequent dynamic NeRF optimization.

In summary, we propose $\boldsymbol{X}$-Planes, a dynamic NeRF pipeline that integrates an effective initialization method and an optimization strategy. By factorizing a 4D tensor derived from a large pretrained model, the initialization method provides a compact and informative representation, along with effective regularization for subsequent dynamic NeRF optimization. Experiments demonstrate the method advantages of rendering performance, convergence speed, and robustness under the few-shot settings. We achieve a PSNR of 20 within 30 seconds, making the optimization $2\times$ faster than the baselines, and the improvements brought by $\boldsymbol{X}$-Planes increases as the number of input views decreases. Additionally, performance improves with larger LRM model sizes, suggesting the potential to benefit from future advancements in LRM. These two key properties open the potential for broader applicability in generalizable dynamic NeRF tasks.

## 2 RELATED WORKS

We first overview two research directions in dynamic NeRF, then briefly discuss Gaussian Splatting for dynamic scene reconstruction. Finally, we review Large Reconstruction Model (LRM) and its related methods.

In this section, we follow the formulation and notation from Yi et al. (2023) and present the standard form of NeRF as:

$$\boldsymbol{p} \xrightarrow{\text{MLP}_\theta} (\boldsymbol{c}, \sigma), \tag{1}$$

where an MLP parameterized by $\theta$ maps spatial positions $\boldsymbol{p} \in \mathbb{R}^3$ to RGB colors $\boldsymbol{c} \in [0,1]^3$ and density $\sigma \in \mathbb{R}_{\geq 0}$.

**Dynamic NeRF: 4D Field.** Representing dynamic objects with NeRF can be achieved in a straightforward manner by conditioning the original NeRF on the timestamp (Park et al., 2021b;a; Li et al., 2022; 2021). This approach treats time $t$ as an additional input dimension, using parameterization method similar to static NeRF:

$$\boldsymbol{p}, t \xrightarrow{\text{MLP}_\theta} (\boldsymbol{c}, \sigma). \tag{2}$$

Subsequent methods (Fridovich-Keil et al., 2023; Shao et al., 2023; Cao and Johnson, 2023) utilize compact feature planes to model the 4D field. These methods, grounded in tensor decomposition theory, significantly accelerate training with less memory usage:

$$\boldsymbol{p}, t \xrightarrow{\text{VT}_\Phi} \boldsymbol{Z} \xrightarrow{\text{MLP}_\theta} (\boldsymbol{c}, \sigma), \tag{3}$$

where the operation VT interpolates the compact feature planes $\Phi$ to produce a latent feature $\boldsymbol{Z} \in \mathbb{R}^d$, which is then decoded into standard radiance field outputs by an MLP with parameters $\theta$. As summarized in Yi et al. (2023), VT typically involves three steps: (i) (Proj) **projecting** input coordinates onto each feature plane, (ii) (Interp$_\Phi$) **interpolating** feature vectors from the corresponding planes $\Phi$, and (iii) (Reduce) **reducing** these interpolated features (e.g., concatenation, multiplication, or addition) to generate the final latent representation $\boldsymbol{Z}$. The process can be formulated as:

$$\boldsymbol{Z} = \text{VT}_\Phi(\boldsymbol{p}, t) = \text{Reduce}(\text{Interp}_\Phi(\text{Proj}(\boldsymbol{p}, t))). \tag{4}$$

*In this paper, we build upon this research direction, further pushing the limits of performance and training speed with large pretrained 3D reconstruction model LRM.*

**Dynamic NeRF: Canonical 3D Field + Motion Field.** To improve the disentanglement of shape and motion, another research line proposes to model the 4D field as a combination of a canonical 3D field and a motion field. The pioneering work (Pumarola et al., 2021) introduces a deformable neural radiance field that adopts a canonical 3D representation with 4D flow fields. Many subsequent methods focus on accelerating static NeRF by using explicit data structures such as feature maps, voxels and tensors. DeVRF (Liu et al., 2022) enables fast non-rigid neural rendering by combining 3D volumetric and 4D voxel fields (Song et al., 2023). V4D (Gan et al., 2024) introduces an effective conditional positional encoding for 4D data, enabling fast novel view synthesis. Additionally, TiNeuVox (Fang et al., 2022b) utilizes optimizable explicit data structures to accelerate radiance field modeling.

**4D Gaussian Splatting.** Some recent works (Xu et al., 2024a; Wu et al., 2024) utilize 3D Gaussian Splatting (3DGS) (Kerbl et al., 2023) to accelerate dynamic scene modeling. Building upon the canonical 3D field with motion field approach, these methods replace the 3D NeRF voxel field with 3DGS to model the canonical 3D field. 3DGS is significantly faster than NeRF under sufficient input views, but struggles with limited input views with challenges such as failed initialization, overfitting on input images, and lack of details as detailed in (Yu et al., 2024). *In real-world dynamic scenes, it is often challenging to capture enough views for a given scene. As a result, research on dynamic NeRF with limited views remains valuable.*

**Large Reconstruction Model (LRM).** The recently proposed LRM (Hong et al., 2024) and its follow-up works (Hong et al., 2024; Li et al., 2024a; Xu et al., 2024b; Wang et al., 2024) leverage a pre-trained vision model, such as DINO (Caron et al., 2021), to encode the input image, and employ a large transformer-based architecture to learn 3D representations of objects from a single image in a data-driven manner. Such architecture consists of two functional components: an encoder $f_\omega$ and a decoder MLP$_\theta$. Their functionalities are formulated as follows:

$$\boldsymbol{I} \xrightarrow{f_\omega} \textcolor{red}{\Phi}, \quad \boldsymbol{p} \xrightarrow{\text{VT}_{\textcolor{red}{\Phi}}} \boldsymbol{Z} \xrightarrow{\text{MLP}_\theta} (\boldsymbol{c}, \sigma), \tag{5}$$

where $f_\omega$ is the encoder that predicts the triplane $\Phi$ from a single image. We can then synthesize any novel views with $\Phi$, the VT operation as defined in Eq.(4), and the generalizable decoder MLP$_\theta$.

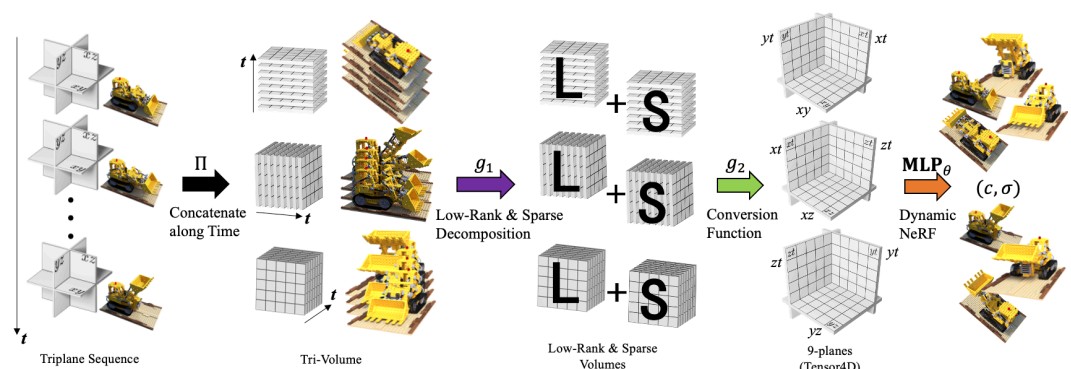

Figure 2: **$X$-Planes pipeline overview.** We omit the LRM $f_\omega$ and start from the output triplane sequence $\{\Phi_t\}_{t=1}^T$. We first reorganize them into three time-aware 4D tensors by chunking and concatenating operator $\Pi$, then get the final feature planes $\Phi$ through low-rank and sparse decomposition $g_1$ and conversion function $g_2$. The feature planes $\Phi$ and LRM's NeRF decoder $\mathbf{MLP}_\theta$ are used to initialize that in dynamic NeRF methods such as $K$-Planes (Fridovich-Keil et al., 2023) and Tensor4D (Shao et al., 2023).

## 3 X-PLANES

In this section, we present $X$-Planes, a pipeline that includes an initialization method followed by a dynamic NeRF optimization strategy. We begin by formulating the pipeline in Sec. 3.1. Next, Sec. 3.2 introduces the low-rank and sparse decomposition strategy, and the reduction technique used in the initialization method. Finally, we discuss the dynamic NeRF optimization strategy in Sec. 3.3.

### 3.1 FORMUTATION

Given a set of images $\{\boldsymbol{I}_1^1, \cdots, \boldsymbol{I}_1^K, \cdots, \boldsymbol{I}_T^1, \cdots, \boldsymbol{I}_T^K\}$ captured over $T$ timestamps of a dynamic scene with $K$ fixed cameras, we select the $k$-th camera as the canonical pose. Using the images $\{\boldsymbol{I}_1^k, \cdots, \boldsymbol{I}_T^k\}$ across all timestamps, a time-series of triplanes $\{\Phi_1, \ldots, \Phi_T\}$ can be predicted using the LRM encoder $f_\omega$, where each triplane $\Phi_t \in \mathbb{R}^{3 \times H \times W \times D}$. The obtained triplanes $\{\Phi_1, \ldots, \Phi_T\}$ form an initial 4D field, which may be noisy and incomplete. Figure 2 illustrates how we decompose and reduce the 4D triplane sequence into compact feature planes. We formally define the entire initialization method of $X$-Planes as follows, where the decomposition and reduction procedure is denoted as $\psi$, with further details provided in section 3.2:

$$\{\boldsymbol{I}_1^k, \ldots, \boldsymbol{I}_T^k\} \xrightarrow{f_\omega} \{\Phi_1, \ldots, \Phi_T\} \xrightarrow{\psi} \Phi. \tag{6}$$

Once we obtain the decomposed feature planes $\Phi$, they are used to initialize the feature plane-based dynamic NeRF methods (Fridovich-Keil et al., 2023; Shao et al., 2023; Cao and Johnson, 2023). These methods can then be efficiently trained following their respective settings with an introduced regularization strategy, formulated as follows:

$$\boldsymbol{p}, t \xrightarrow{\text{VT}_\Phi} \boldsymbol{Z} \xrightarrow{\text{MLP}_\theta} (\boldsymbol{c}, \sigma). \tag{7}$$

### 3.2 TRIPLANE SEQUENCE TO COMPACT FEATURE PLANES

As illustrated in Figure 2, given a time-series sequence of triplanes $\{\Phi_1, \ldots, \Phi_T\} \in \mathbb{R}^{3 \times T \times H \times W \times D}$ obtained from LRM, we first reorganize them into three time-aware 4D tensors by chunking along the spatial dimension and concatenating along the time dimension:

$$\Pi(\Phi_1, \ldots, \Phi_T) = \{\Phi_{xyt}, \Phi_{yzt}, \Phi_{xzt}\}, \tag{8}$$

where $\Phi_{xyt}, \Phi_{yzt}, \Phi_{xzt} \in \mathbb{R}^{T \times H \times W \times D}$, and $\Pi$ is the composite of chunking and concatenating operations. The three time-aware 4D tensors can be regarded as three distinct views in the feature space, stacked along the time axis. Our objective is to derive the final feature planes

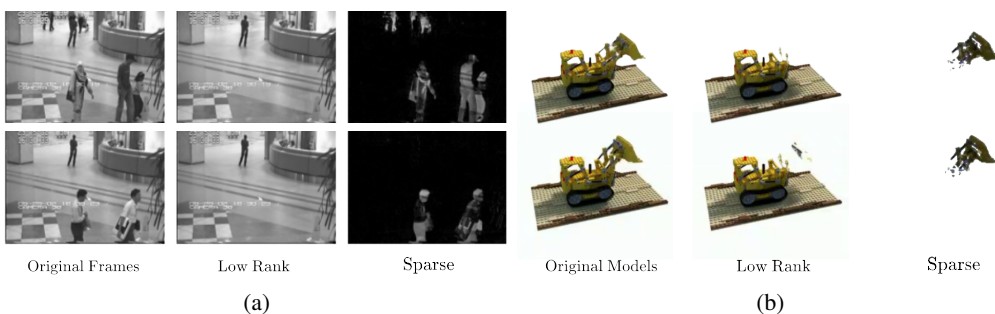

Original Frames    Low Rank    Sparse     Original Models    Low Rank    Sparse

(a)             (b)

Figure 3: **Low-rank & sparse decomposition examples.** (a) Background and static objects are reserved in the low-rank, and moving pedestrians are extracted in the sparse. (b) The static components of the bulldozer remain in the low-rank, and the moving shovel are decomposed into the sparse.

$\mathbf{\Phi} = \{\cdots, \boldsymbol{P}_{xy}, \boldsymbol{P}_{yz}, \boldsymbol{P}_{xz}, \cdots\}$ ($\boldsymbol{P}_{\{.\}} \in \mathbb{R}^{H \times W \times D}$) through low-rank and sparse decomposition function $g_1$ and a conversion function $g_2$ which transforms the decomposed components into optimizable feature planes. The overall transformation is represented as $\psi = g_2 \circ g_1$. Taking the 9 feature plane representation from Tensor4D (Shao et al., 2023) as an example, the process can be summarized as follows:

$$\boldsymbol{\Phi}_{xyt} \xrightarrow{g_1} \{\hat{\boldsymbol{L}}_{xyt}, \hat{\boldsymbol{S}}_{xyt}\} \xrightarrow{g_2} \{\boldsymbol{P}_{xy}, \boldsymbol{P}_{xt}, \boldsymbol{P}_{yt}\}$$
$$\boldsymbol{\Phi}_{yzt} \xrightarrow{g_1} \{\hat{\boldsymbol{L}}_{yzt}, \hat{\boldsymbol{S}}_{yzt}\} \xrightarrow{g_2} \{\boldsymbol{P}_{yz}, \boldsymbol{P}_{yt}, \boldsymbol{P}_{zt}\} \qquad (9)$$
$$\boldsymbol{\Phi}_{xzt} \xrightarrow{g_1} \{\hat{\boldsymbol{L}}_{xzt}, \hat{\boldsymbol{S}}_{xzt}\} \xrightarrow{g_2} \{\boldsymbol{P}_{xz}, \boldsymbol{P}_{xt}, \boldsymbol{P}_{zt}\}$$

The 6-plane representation from K-Plane (Fridovich-Keil et al., 2023) can be seen as a special case of the 9-plane representation, where feature planes with the same subscripts share parameters. The respective advantages and disadvantages of the 6-plane and 9-plane representations are discussed in the original papers.

**Low-Rank and Sparse Decomposition $g_1$.** The low rank and sparse decomposition approach is motivated by the successful application of robust PCA in video surveillance (Wright and Ma, 2022). Given a sequence of surveillance video frames, we often aim to identify dynamic activities against a static background. Stacking video frames into a matrix $\boldsymbol{Y} \in \mathbb{R}^{(H \times W) \times T}$, the low-rank component $\boldsymbol{L} \in \mathbb{R}^{(H \times W) \times T}$ represents the stationary background, while the sparse component $\boldsymbol{S} \in \mathbb{R}^{(H \times W) \times T}$ captures the moving objects in the foreground. The decomposed low-rank component $\hat{\boldsymbol{L}}$ and sparse component $\hat{\boldsymbol{S}}$ can be obtained by solving the following optimization problem:

$$\begin{aligned} \min \quad & \|\boldsymbol{L}\|_F + \lambda\|\boldsymbol{S}\|_1 \\ \text{s.t} \quad & \boldsymbol{L} + \boldsymbol{S} = \boldsymbol{Y}, \end{aligned} \qquad (10)$$

where $\|\boldsymbol{L}\|_F$ denotes the Frobenius norm, promoting low-rank structure in $\boldsymbol{L}$, and $\|\boldsymbol{S}\|_1$ is the $l_1$-norm encouraging sparsity in $\boldsymbol{S}$. The parameter $\lambda$ balances the contribution of the low-rank and sparse components. This optimization can be efficiently solved using the Alternating Directions Method of Multipliers (ADMM) (Wright and Ma, 2022). The detailed procedure can be found in Appendix section. A specific example and its decomposition results are illustrated in Figure 3.

To apply robust PCA to the three time-aware 4D tensors in Eq.(8), we first reshape them to matrices $\hat{\boldsymbol{\Phi}}_{xyt}, \hat{\boldsymbol{\Phi}}_{yzt}, \hat{\boldsymbol{\Phi}}_{xzt} \in \mathbb{R}^{(H \times W \times D) \times T1}$. The reformulated optimization objective, adapted from Eq.(10), is expressed as:

$$\begin{aligned} \min \quad & \|\boldsymbol{L}\|_F + \lambda\|\boldsymbol{S}\|_1 \\ \text{s.t} \quad & \boldsymbol{L} + \boldsymbol{S} = \hat{\boldsymbol{\Phi}}_c, \quad c \in \{xyt, yzt, xzt\}. \end{aligned} \qquad (11)$$

The final Lagrange function, formulated using the method of Lagrange multiplier, is:

$$\operatorname{argmin}_{\boldsymbol{L},\boldsymbol{S}} \quad \|\boldsymbol{L}\|_F + \lambda\|\boldsymbol{S}\|_1 + \frac{\mu}{2}\|\boldsymbol{L} + \boldsymbol{S} - \hat{\boldsymbol{\Phi}}\|_2^2, \qquad (12)$$

---

[1]To avoid ambiguity, all variable with $\hat{\cdot}$ ($\hat{\boldsymbol{\Phi}}, \hat{\boldsymbol{L}}, \hat{\boldsymbol{S}}$) indicate 2D matrics and $\tilde{\cdot}$ ($\tilde{\boldsymbol{\Phi}}, \tilde{\boldsymbol{L}}, \tilde{\boldsymbol{S}}$) indicates 4D tensors.

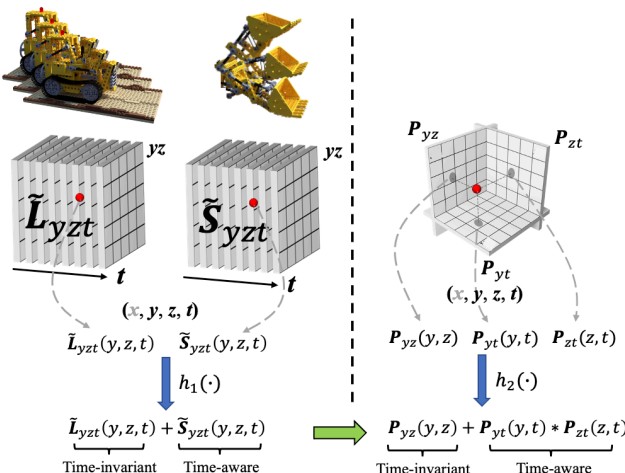

Figure 4: **Illustration of conversion function $g_2$.** Taking $(y, z, t)$ as an example, the conversion function factorizes the feature voxels of the low-rank and sparse into planes.

which can be solved efficiently using ADMM. By applying $\hat{\boldsymbol{\Phi}}_{xyt}, \hat{\boldsymbol{\Phi}}_{yzt}, \hat{\boldsymbol{\Phi}}_{xzt}$ to Eq.(12) independently, we obtain three pairs of low-rank and sparse matrices $(\hat{\boldsymbol{L}}_{xyt}, \hat{\boldsymbol{S}}_{xyt})$, $(\hat{\boldsymbol{L}}_{yzt}, \hat{\boldsymbol{S}}_{yzt})$, $(\hat{\boldsymbol{L}}_{xzt},$ $\hat{\boldsymbol{S}}_{xzt})$. Each of these 6 matrices has the same dimension, $\mathbb{R}^{(H \times W \times D) \times T}$. To facilitate subsequent mathematical derivations, these matrices are reshaped back into 4D tensors: $(\tilde{\boldsymbol{L}}_{xyt}, \tilde{\boldsymbol{S}}_{xyt})$, $(\tilde{\boldsymbol{L}}_{yzt},$ $\tilde{\boldsymbol{S}}_{yzt})$, $(\tilde{\boldsymbol{L}}_{xzt}, \tilde{\boldsymbol{S}}_{xzt})$, with dimensions $\mathbb{R}^{H \times W \times T \times D}$.

**Conversion function $g_2$.**  In this paragraph, we discuss how to transform the decomposed low-rank and sparse components into optimizable feature planes. Before that, we must clarify the connection between the decomposed low-rank and sparse 4D tensors and compact feature planes.

As shown in Figure 4, suppose we have a pair of low-rank & sparse 4D tensors and 3 feature planes, sharing the same decoder. Both of them can utilize this decoder to render an image at arbitrary viewpoints and timestamps. To render a given point $(x, y, z)$ at the $t$-th timestamp, we first need to sample the corresponding features: feature pair $\{\tilde{\boldsymbol{L}}_{yzt}(y, z, t), \tilde{\boldsymbol{S}}_{yzt}(y, z, t)\}$ from the low-rank & sparse 4D tensors, and triplet features $\{\boldsymbol{P}_{yz}(y, z), \boldsymbol{P}_{yt}(y, t), \boldsymbol{P}_{zt}(z, t)\}$ from the 3 feature planes. The key challenge is to determine the combination functions, $h_1(\cdot)$ and $h_2(\cdot)$, that satisfy the subsequent equations:

$$h_1(\tilde{\boldsymbol{L}}_{yzt}(y, z, t), \tilde{\boldsymbol{S}}_{yzt}(y, z, t)) = h_2(\boldsymbol{P}_{yz}(y, z), \boldsymbol{P}_{yt}(y, t), \boldsymbol{P}_{zt}(z, t)). \tag{13}$$

The second assumption is that the shared decoder is identical to the NeRF decoder used in LRM. Under this assumption, $h_1(\cdot)$ can be defined as an additive operation, following the constraint from Eq.(11), which ensures that the low-rank and sparse matrices sum back to the original feature $\hat{\boldsymbol{\Phi}}$ which can be directly rendered by the LRM's NeRF decoder. Furthermore, as illustrated in Figure 3, the low-rank component is time-invariant, implying that the sparse component must be time-aware. Consequently, $h_2(\cdot)$ also becomes an additive operator that combines time-invariant and time-aware components. In the triplet features $\{\boldsymbol{P}_{yz}(y, z), \boldsymbol{P}_{yt}(y, t), \boldsymbol{P}_{zt}(z, t)\}$, only $\boldsymbol{P}_{yz}(y, z)$ is time-invariant, while $\boldsymbol{P}_{yt}(y, t), \boldsymbol{P}_{zt}(z, t)$ are time-aware. Following the approach from (Fridovich-Keil et al., 2023), we multiply the time-aware components to obtain the desired features. This leads to the final forms of $h_1(\cdot)$ and $h_2(\cdot)$, as depicted in Figure 4 [2]. Finally, we adopt the feature merging strategy of LRM for sampled features from the planes. These features are concatenated and subsequently fed into the MLP decoder. This process yields the concatenated low-rank and sparse features for $(x, y, z, t)$:

$$\begin{bmatrix} \boldsymbol{\Phi}_{xyt}(x, y, t) \\ \boldsymbol{\Phi}_{yzt}(y, z, t) \\ \boldsymbol{\Phi}_{xzt}(x, z, t) \end{bmatrix} = \begin{bmatrix} \tilde{\boldsymbol{L}}_{xyt}(x, y, t) \\ \tilde{\boldsymbol{L}}_{yzt}(y, z, t) \\ \tilde{\boldsymbol{L}}_{xzt}(x, z, t) \end{bmatrix} + \begin{bmatrix} \tilde{\boldsymbol{S}}_{xyt}(x, y, t) \\ \tilde{\boldsymbol{S}}_{yzt}(y, z, t) \\ \tilde{\boldsymbol{S}}_{xzt}(x, z, t) \end{bmatrix}, \tag{14}$$

---

[2]For simplicity, we take the $yzt$ part as an example. The $xyt$ and $xzt$ parts share symmetric operations.

and the corresponding features from feature planes:

$$
\begin{bmatrix} \boldsymbol{P}_{xy}(x,y) \\ \boldsymbol{P}_{yz}(y,z) \\ \boldsymbol{P}_{xz}(x,z) \end{bmatrix} + \begin{bmatrix} \boldsymbol{P}_{xt}(x,t) \cdot \boldsymbol{P}_{yt}(y,t) \\ \boldsymbol{P}_{yt}(y,t) \cdot \boldsymbol{P}_{zt}(z,t) \\ \boldsymbol{P}_{xt}(x,t) \cdot \boldsymbol{P}_{zt}(z,t) \end{bmatrix}. \tag{15}
$$

Straightforwardly, we can use the right side of Eq.(14) to supervise Eq.(15) during dynamic NeRF training. However, as shown in Figure 3, the decomposed low-rank matrices hold the time-invariant information, satisfying the requirements for $\boldsymbol{P}_{xy}, \boldsymbol{P}_{yz}, \boldsymbol{P}_{xz}$ in Eq.(9). From a visual perspective, the decomposed low-rank matrices $\hat{\boldsymbol{L}}$ is indistinguishable along the time axis. Therefore, we can simply choose the chunk of $\hat{\boldsymbol{L}}$ corresponding to the $\frac{T}{2}$ timestamp to **initialize** the time-invariant feature planes:

$$
\boldsymbol{P}_{xy} \doteq \tilde{\boldsymbol{L}}_{xyt}\left(\frac{T}{2}\right), \quad \boldsymbol{P}_{yz} \doteq \tilde{\boldsymbol{L}}_{yzt}\left(\frac{T}{2}\right), \quad \boldsymbol{P}_{xz} \doteq \tilde{\boldsymbol{L}}_{xzt}\left(\frac{T}{2}\right), \tag{16}
$$

where $\tilde{\boldsymbol{L}}_{xyt}(\frac{T}{2})$ indicates selection along the time axis of the 4D tensor $\tilde{\boldsymbol{L}}_{xyt}$ corresponding to the $\frac{T}{2}$-th timestamp. The ablation study in the experiment section shows that the average rank of low-rank matrics is 2.53 which also aligns with the conclusions of our analysis.

### 3.3 Dynamic NeRF Optimization

**Feature Decoders MLP$_\theta$.** We use the pretrained NeRF MLP decoder from LRM as our feature decoder. The MLP is comprised of multiple linear layers with ReLU activations. The output is a 4-dimensional vector, where the first three dimensions represent the RGB colors, and the last dimension corresponds to the density of the field.

**Losses.** In addition to the regularization terms introduced in (Fridovich-Keil et al., 2023; Shao et al., 2023), we incorporate the sparse component $\tilde{\boldsymbol{S}}$ to regularize the time-aware feature planes $\boldsymbol{P}_S$ with loss:

$$
\mathcal{L}_r = \|\tilde{\boldsymbol{S}} - \boldsymbol{P}_S\|_2^2, \tag{17}
$$

where $\tilde{\boldsymbol{S}}$ is the last term in Eq.(14) and $\boldsymbol{P}_S$ is the second term in Eq.(15), namely:

$$
\tilde{\boldsymbol{S}} = \begin{bmatrix} \tilde{\boldsymbol{S}}_{xyt}(x,y,t) \\ \tilde{\boldsymbol{S}}_{yzt}(y,z,t) \\ \tilde{\boldsymbol{S}}_{xzt}(x,z,t) \end{bmatrix}, \quad \boldsymbol{P}_S = \begin{bmatrix} \boldsymbol{P}_{xt}(x,t) * \boldsymbol{P}_{yt}(y,t) \\ \boldsymbol{P}_{yt}(y,t) * \boldsymbol{P}_{zt}(z,t) \\ \boldsymbol{P}_{xt}(x,t) * \boldsymbol{P}_{zt}(z,t) \end{bmatrix}.
$$

## 4 Experiments

**Dataset.** The D-NeRF dataset (Pumarola et al., 2021) contains eight scenes of varying duration (50 to 200 frames). Each timestamp provides a single training image from a different viewpoint. In our setting, we require a monocular video with a fixed viewpoint as input to the LRM to generate the initial triplanes. Hence, we train K-Planes for the eight scenes and render a time-series of fixed-view images. For evaluation, standardized test views are taken from novel camera positions at various timestamps throughout the video.

**Baselines.** We compare our method against the state-of-the-art (SOTA) from two research directions in dynamic NeRF, Tensor4D (Shao et al., 2023) and K-Planes (Fridovich-Keil et al., 2023), which directly learn time-conditioned 4D fields. For fair comparison, we retrain them in our experiments, achieving improved results compared to their original papers. On the other hand, TiNeuVox (Fang et al., 2022b) is the SOTA methods from the other direction: modelling a canonical 3D field with an associated motion field. Both methods rely on explicit voxel grids to represent the canonical 3D field. Apart from the NeRF-based methods, we also compare our method with the gaussian splatting methods, including 4D-GS (Wu et al., 2024), Spacetime GS (Li et al., 2024b), and Saro-GS (Yan et al., 2024).

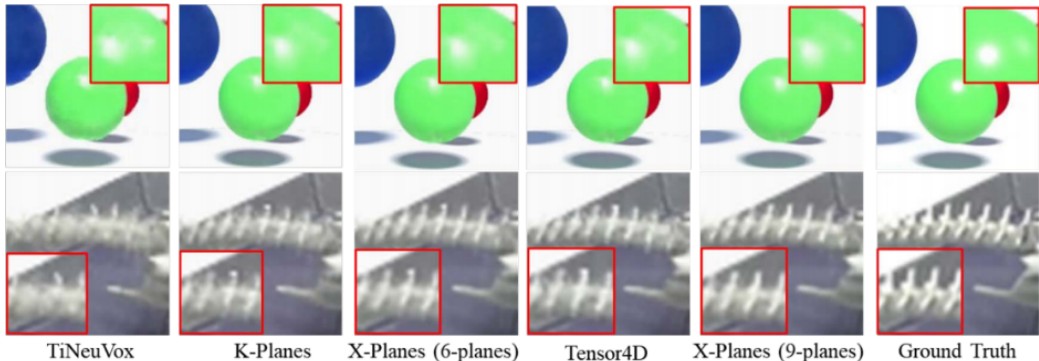

Figure 5: **Qualitative comparison with SOTA methods.**

**Implementation Details.** For robust PCA, we set $\mu = 0.05$ and $\lambda = 0.009$ for the ADMM optimization. We borrow the LRM architecture as the same as that in the openLRM Github repository. In our experiments, we evaluate $X$-Planes with two configurations: 6-planes and 9-planes. The ablation study on different LRM versions of varying parameter sizes, multi-resolution configurations, and the 6/9-plane representations of $X$-Planes is provided in the Appendix section. All the experiments are trained on a single NVIDIA 3090 GPU.

Table 1: **Quantitative comparison on D-NeRF dataset.**

| Genre | Name | **PSNR↑** | **SSIM↓** | **Time ↓** |
|-------|------|-----------|-----------|------------|
| | 4D-GS | 34.09 | 0.981 | 24 min |
| GS | Spacetime GS | 36.07 | 0.984 | 50 min |
| | Saro-GS | 36.13 | 0.985 | 50 min |
| | $K$-Planes | 31.71 | 0.971 | 52 min |
| | Tensor4D | 32.33 | 0.973 | 64 min |
| NeRF | TiNeuVox | 32.67 | 0.973 | 28 min |
| | $X$-Planes (9 Planes) | 33.67 | 0.980 | 34 min |
| | $X$-Planes (6 Planes) | 33.40 | 0.979 | 26 min |

Table 2: **Quantitative comparison on different LRM sizes.**

| **Size** | **PSNR↑** | **SSIM↓** |
|----------|-----------|-----------|
| Small | 31.13 | 0.966 |
| Base | 31.78 | 0.970 |
| Large | 32.18 | 0.973 |
| xLarge | 33.40 | 0.979 |

### 4.1 PERFORMANCE COMPARISON

**Comparison on reconstruction accuracy.** We train our model for each individual scene and provide example results for novel view synthesis in Figure.5. The quantitative results in Table 1 demonstrates that our method outperforms NeRF-based SOTA methods in both rendering quality and reconstruction speed. With $2\times$ training speed, $X$-Planes achieves 1.21 PSNR improvement over the SOTA method Tensor4D, and 2.7 PSNR over $K$-Planes using similar 6-plane representation. $X$-Planes also produce competitive results compared with gaussian splatting methods.

**Comparison on few training iterations.** We further compare $X$-Planes with baseline methods under a limited number of training iterations to highlight the effectiveness of our $X$-Planes initialization method. As shown in Figure 6, $X$-Planes achieves relatively good visual results with just 200 optimization iterations. This demonstrates the potential of $X$-Planes for fast deployment in applications where speed is critical and slight compromises in quality are acceptable.

### 4.2 FEW-SHOT AND DIFFERENT LRM MODELS

**Effects of different LRM model sizes.** In an ideal scenario, the LRM can perfectly reconstruct a static scene. Using our $X$-Planes initialization method, this enables the direct construction of a compact 4D field with feature planes, capable of rendering high-quality images at arbitrary viewpoints and timestamps. Consequently, the effectiveness and efficiency of $X$-Planes should improve as LRM

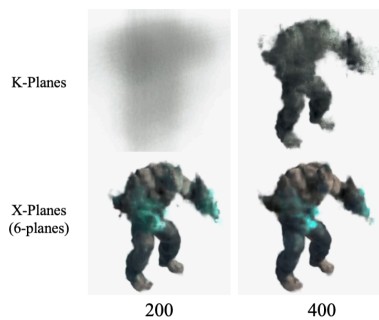

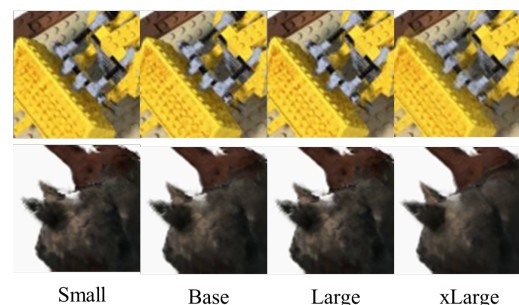

Figure 6: **Comparison with random initialization**. $X$-Planes' training converges faster than the vanilla. Bottom numbers indicate iterations.

Figure 7: **Qualitative comparison on different model size of LRM.** As the LRM size increases, the reconstruction result quality improves.

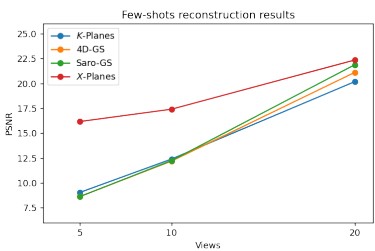

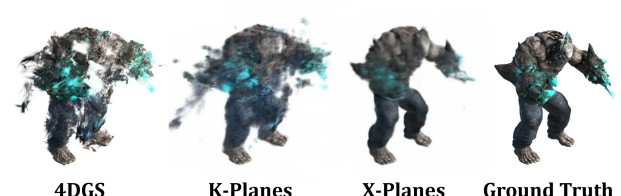

Figure 8: **Quantitative comparison of few-shots reconstruction.**

Figure 9: **Qualitative comparison of models under few-shots settings.** The number of input views is 5.

performance increases, highlighting the importance of an effective initialization strategy. Table 2 and Figure 7 provide both quantitative and qualitative evidences showing that $X$-Planes benefits from enhanced LRM performance, where LRM models of varying sizes are used to approximate models with varying capabilities. We borrow different LRM model sizes from Github Repositories[34]. Small, Base, and xLarge in Table.2 correspond to the models small, base, and large from the OpenLRM repository. Large in Table 2 refers to the model from the TripoSR repository.

**Few-shot results.** Figure 8 shows the PSNR and qualitative results of $K$-Planes, SaRO-GS (Yan et al., 2024), 4DGS (Wu et al., 2024) and $X$-Planes at different few-shot levels (5 views, 10 views, and 20 views). Figure 9 shows the corresponding visualized results. As the # of views decreases, the advantage of $X$-Planes over the other methods increases. Experiments prove that given a few input images, the initialization and regularization of the LRM-generated planes make the model more robust to overfitting. This result, cooperates with that in different LRM model sizes, opening the potential for broader applicability in generalizable dynamic NeRF tasks.

## 5 CONCLUSION

This paper proposes $X$-Planes, a novel dynamic NeRF pipeline with an effective initialization and optimization strategy. It employs a pretrained Large Reconstruction Model (LRM) to generate an initial noisy and incomplete 4D field of a dynamic scene. This initial representation is subsequently decomposed into compact feature planes through low-rank and sparse decomposition. The resulting decomposed feature planes serve as both an effective initialization and a form of regularization for subsequent dynamic NeRF optimization, enabling new state-of-the-art results demonstrated through qualitative and quantitative results. Additionally, $X$-Planes empowers fast deployment in applications where speed is prioritized over absolute quality. Furthermore, $X$-Planes is broadly applicable to dynamic NeRF methods, ready to benefit from future advancements of LRM, paving the way for more generalizable dynamic NeRF tasks.

---

[3]https://github.com/3DTopia/OpenLRM
[4]https://github.com/VAST-AI-Research/TripoSR

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

# APPENDICES

## ALTERNATING DIRECTIONS METHOD OF MULTIPLIERS (ADMM)

In this section, we concretize how the low-rank and sparse decomposition, i.e., the robust PCA, is achieved by the Alternating Directions Method of Multipliers (ADMM) algorithm, which solves the optimization objective below,

$$\min \quad \|\boldsymbol{L}\|_F + \lambda\|\boldsymbol{S}\|_1$$
$$\text{s.t} \quad \boldsymbol{L} + \boldsymbol{S} = \hat{\boldsymbol{\Phi}}, \tag{18}$$

where $\hat{\boldsymbol{\Phi}}$ is the known feature matrix (refer to Eq.(18) the of the main body), and $\boldsymbol{L}$ and $\boldsymbol{S}$ are the unknown low-rank and sparse components, respectively. A generic Lagrange multiplier algorithm would optimize the augmented Lagrangian, adapted from Eq.(19),

$$\mathcal{L}_\mu(\boldsymbol{L}, \boldsymbol{S}, \boldsymbol{\Lambda}) = \|\boldsymbol{L}\|_F + \lambda\|\boldsymbol{S}\|_1 + \langle\boldsymbol{\Lambda}, \boldsymbol{\Delta}\rangle + \frac{\mu}{2}\|\boldsymbol{\Delta}\|_2^2, \tag{19}$$

where $\langle\cdot, \cdot\rangle$ denotes the matrix inner product, and $\boldsymbol{\Delta}$ denotes the residual matrix term,

$$\boldsymbol{\Delta} = \boldsymbol{L} + \boldsymbol{S} - \hat{\boldsymbol{\Phi}}. \tag{20}$$

With the lagrange multiplier $\boldsymbol{\Lambda}$ initialized as $\boldsymbol{0}$, the Lagrange multiplier algorithm would repeatedly update

$$(\boldsymbol{L}_{k+1}, \boldsymbol{S}_{k+1}) = \arg\min_{\boldsymbol{L}, \boldsymbol{S}} \mathcal{L}_\mu(\boldsymbol{L}, \boldsymbol{S}, \boldsymbol{\Lambda}_k),$$
$$\boldsymbol{\Lambda}_{k+1} = \boldsymbol{\Lambda}_k + \mu\boldsymbol{\Delta}_{k+1}. \tag{21}$$

To efficiently solve the optimization problem above, the ADMM algorithm recognizes it as two sub-problems: $\min_{\boldsymbol{L}} \mathcal{L}_\mu(\boldsymbol{L}, \boldsymbol{S}, \boldsymbol{\Lambda})$ and $\min_{\boldsymbol{S}} \mathcal{L}_\mu(\boldsymbol{L}, \boldsymbol{S}, \boldsymbol{\Lambda})$.

The sparse component is minimized by

$$\arg\min_{\boldsymbol{S}} \mathcal{L}_\mu(\boldsymbol{L}, \boldsymbol{S}, \boldsymbol{\Lambda}) = \mathcal{S}_{\lambda/\mu}(\hat{\boldsymbol{\Phi}} - \boldsymbol{L} - \mu^{-1}\boldsymbol{\Lambda}), \tag{22}$$

where $\mathcal{S}_{\lambda/\mu}$ denotes the shrinkage operator. For a single element $x$ in the matrix,

$$\mathcal{S}_{\lambda/\mu}[x] = \text{sgn}(x)\max(|x| - \lambda/\mu, 0), \tag{23}$$

and the operation extends to matrices in an element-wise way.

The low-rank component is minimized by

$$\arg\min_{\boldsymbol{L}} \mathcal{L}_\mu(\boldsymbol{L}, \boldsymbol{S}, \boldsymbol{\Lambda}) = \mathcal{D}_{1/\mu}(\hat{\boldsymbol{\Phi}} - \boldsymbol{S} - \mu^{-1}\boldsymbol{\Lambda}). \tag{24}$$

Here, $\mathcal{D}_{1/\mu}$ denotes the singular value thresholding operator,

$$\mathcal{D}_{1/\mu}(\boldsymbol{M}) = \boldsymbol{U}\mathcal{S}_{1/\mu}(\boldsymbol{\Sigma})\boldsymbol{V}^*, \tag{25}$$

where $\boldsymbol{M} = \boldsymbol{U}\boldsymbol{\Sigma}\boldsymbol{V}^*$ is the singular value decomposition.

The strategy of the ADMM algorithm is to first optimize $\mathcal{L}_\mu$ with respect to $\boldsymbol{L}$ (fixing $\boldsymbol{S}$), then minimize $\mathcal{L}_\mu$ with respect to $\boldsymbol{S}$ (fixing $\boldsymbol{L}$), and then finally update the Lagrange multiplier $\boldsymbol{\Lambda}$. The process is summarized as **Algorithm** 1.

---
**Algorithm 1** Robust PCA by ADMM

---
1: Initialize: $\boldsymbol{S}_0 = \boldsymbol{\Lambda}_0 = 0, \mu > 0$.
2: **while** not converged **do**
3:  Compute $\boldsymbol{L}_{k+1} = \mathcal{D}_{1/\mu}(\hat{\boldsymbol{\Phi}} - \boldsymbol{S}_k - \mu^{-1}\boldsymbol{\Lambda}_k)$;
4:  Compute $\boldsymbol{S}_{k+1} = \mathcal{S}_{\lambda/\mu}(\hat{\boldsymbol{\Phi}} - \boldsymbol{L}_{k+1} - \mu^{-1}\boldsymbol{\Lambda}_k)$;
5:  Compute $\boldsymbol{\Lambda}_{k+1} = \boldsymbol{\Lambda}_k + \mu(\boldsymbol{L}_{k+1} + \boldsymbol{S}_{k+1} - \hat{\boldsymbol{\Phi}})$;
6: **end while**
7: Output: $\boldsymbol{L}_\star \leftarrow \boldsymbol{L}_k; \boldsymbol{S}_\star \leftarrow \boldsymbol{S}_k$.

---

ABLATION EXPERIMENTS ON LRM SIZES, RESOLUTIONS, AND NUMBERS OF PLANES

In this section, we experiment with our $X$-Planes on different LRM model sizes, multi-resolution, and 6/9-planes representations. With such ablation studies, we investigate the performance gains from various aspects of $X$-Planes respectively.

**LRM size and 6/9-planes representation.** In Table. 3, we ablate our model with respect to different sizes of the LRM models and the number of feature planes. It shows the larger model size of LRM would directly increase the performance of $X$-Planes.

Table 3: **Ablation study over LRM sizes and six/nine-planes representation.**

| Model Size | 6-Planes PSNR ↑ | 9-Planes PSNR ↑ | 6-Planes SSIM ↑ | 9-Planes SSIM ↑ |
|---|---|---|---|---|
| Small | 31.13 | 32.04 | 0.969 | 0.972 |
| Base | 31.78 | 32.54 | 0.971 | 0.975 |
| Large | 32.18 | 32.54 | 0.972 | 0.974 |
| xLarge | 33.40 | 33.67 | 0.978 | 0.981 |

**Multi-resolution.** In Table 4, we use LRM model with xLarge size and ablate our model with respect to different resolutions of the feature plane and their impact on performance metrics. The results refer to a different conclusion from that in Tensor4D and K-Planes. In Tensor4D and K-Planes, the performance increases significantly with the increase of feature plane resolution, whereas it is not influenced much by feature plane resolution in our method. The MLP decoder architecture and its pretrained weights (from LRM) might be the main reason. We will explore it in our feature work.

Table 4: **Ablation study over plane resolutions.**

| Plane Resolution ↑ | 6-Planes PSNR ↑ | 9-Planes PSNR ↑ | 6-Planes SSIM ↑ | 9-Planes SSIM ↑ |
|---|---|---|---|---|
| 64 | 33.40 | 33.56 | 0.979 | 0.980 |
| 128 | 33.51 | 33.65 | 0.980 | 0.981 |
| 256 | 33.55 | 33.68 | 0.981 | 0.981 |
| 512 | 33.55 | 33.67 | 0.981 | 0.981 |

MORE QUALITATIVE RESULTS

This section shows more cases of Section 4.1. (see Figure 10)

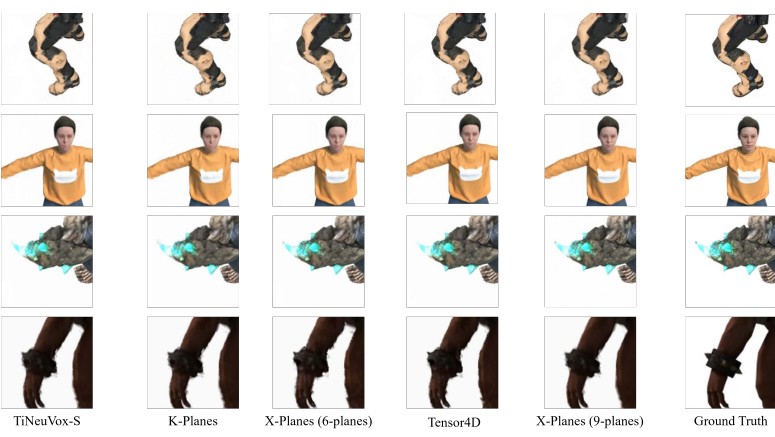

| TiNeuVox-S | K-Planes | X-Planes (6-planes) | Tensor4D | X-Planes (9-planes) | Ground Truth |

Figure 10: Qualitative comparison with SOTA methods.

