# OpenReview forum: "$X$-Planes: Adaptive and Efficient Representation for Dynamic Reconstruction and Rendering in the Age of Large Pretrained Models"
_ICLR.cc/2026/Conference — ICLR 2026 Conference Withdrawn Submission_

### Official Review · Reviewer_54qX · 2025-10-27

**Soundness:** 2
**Presentation:** 2
**Contribution:** 1
**Rating:** 2
**Confidence:** 4

**Summary:**

This paper introduces a dynamic NeRF framework that leverages a Large Reconstruction Model (LRM) for initialization. Specifically, given a set of multi-view videos, the proposed method first uses a pretrained LRM to produce a sequence of triplanes for each video frame. These triplanes form an initial 4D field, which is then decomposed via Robust PCA into low-rank and sparse components. The decomposed components are converted into compact 6- or 9-plane feature representations that serve as the initialization for a downstream K-planes training. The LRM’s pre-trained decoder is also reused to initialize the NeRF MLP.
The authors claim faster convergence, higher PSNR, and improved few-shot robustness compared to baseline NeRF methods, though still lagging behind recent Gaussian Splatting–based methods in both accuracy and efficiency.

**Strengths:**

* Thanks to the initialization from the pre-trained LRM, the proposed method produces better results than baselines under few-shot settings.

* The paper includes ablations on LRM model size and shows consistent trends, suggesting the potential to benefit from future advancements in LRM.

**Weaknesses:**

* **Lack of video results for dynamic reconstruction.**
While static reconstructions and still images are shown, the paper does not include dynamic video comparisons in the supplemental materials (e.g., rendering quality over time). This omission makes it difficult to assess temporal consistency, which is a key aspect of dynamic NeRFs.


* **Cannot outperform GS-based methods.**
Despite claiming efficiency, the proposed approach remains significantly slower and less accurate than Gaussian Splatting (GS) variants such as 4D-GS, Spacetime GS, and Saro-GS. Table 1 shows that X-Planes achieves PSNR 33.7 vs 36.1 for Saro-GS, and training still takes 26–34 minutes per scene. Compared to 4DGS, the proposed method is both slower and less accurate.


* **Weak novelty.**
The main idea of using pretrained LRM outputs as initialization is conceptually straightforward and arguably an engineering choice rather than a novel algorithmic contribution. The contribution lies mainly in integrating these existing components.


* **Limited efficiency gains.**
Although X-Planes converges faster than K-Planes/Tensor4D, it seems to remain far from real-time rendering. In contrast, GS-based methods naturally support real-time rendering once trained. In addition, the training efficiency improvements (≈2×) are modest compared to recent acceleration techniques (e.g., GS, TiNeuVox, voxel-based NeRFs).


* **Dependence on external large models.**
The approach heavily relies on the heavy pretrained LRM, yet the performance gains enabled by this additional prior appear limited.. What's worse, this dependence may limit generalization to scenes outside LRM’s domain. In contrast, baseline methods such as 4DGS and Tensor4D are trained from scratch and still achieve comparable, if not even superior, results without such a heavy dependency. This raises questions regarding the necessity and justification for incorporating the pretrained LRM within the proposed framework.

**Questions:**

See [weaknesses].

---

### Official Review · Reviewer_xWvS · 2025-10-31

**Soundness:** 1
**Presentation:** 1
**Contribution:** 1
**Rating:** 2
**Confidence:** 4

**Summary:**

This paper presents a framework for learning factorization-based dynamic representation by taking as initialization the Tri-Planes output of a pre-trained Large Reconstruction Model (LRM).
This output is then factorized into low-rank volumes that largely model the static parts of a scene, and sparse volumes that model the dynamic parts of a scene.
Subsequently, a conversion function is applied on the low-rank and sparse volumes to obtain well-established representations such as Tensor4D and K-Planes.
The authors claim that their method is both faster and of higher quality and Tensor4D and K-Planes.

**Strengths:**

- The ideas presented in this paper are original for factorization-based 3D representations, and are grounded in other parts of the literature.

**Weaknesses:**

## (w.1) Discrepancy between the method and its evaluation protocol

My major concern is the discrepancy between the nature of the method and its evaluation protocol.
This paper evokes two lines of works in the literature:
- **Feed-forward 3D modeling**, i.e. large models that have been pre-trained on large datasets in order to infer the parameters of NeRF-based representations (e.g. Tri-Planes) from images. These models typically have an extremely costly pre-training procedure, but an inference time that is relatively fast compared to directly optimizing a 3D representation.
    - For instance, the LRM model evoked and utilized in this paper is a model with 500 million parameters, trained for about 3 days on 128 NVIDIA A100 GPUs. Subsequent to this training, LRM can infer a 3D representation in less than 5 seconds on a single NVIDIA A100 GPU.
- **NeRF-based scene representations**, i.e., scene representations that are trained to model a 3D scene from its captured images (e.g. NeRF, Tri-Planes, K-Planes, 3DGS). These models typically have a relatively reasonable training time: less than an hour for Tri-Planes, K-Planes and 3DGS. Once trained, such representations can be used for Novel View Synthesis (NVS).

The work presented in this paper utilizes LRM, a pre-trained feed-forward model, to infer a 3D representations.
After this inference, a training pipeline is applied on its output in order to improve its quality.
As such, this method improves upon feed-forward modeling (the first line of work evoked above), by adding a "post-processing" (or post-training, fine-tuning) step to the output of feed-forward models.

The authors claim that LRM followed by their pipeline results in representations that of higher quality than NeRF-based scene representations trained from scratch.
Additionally, the authors claim that their method, i.e. inferring an LRM followed by a post-processing, is faster than training NeRF-based scene representations such as K-Planes.
Moreover, the authors present in section 4.1 a comparison on "few training iterations", where they compare the quality obtained after a few iterations of "post-processing" an inferred scene representation, with the quality obtained after a few iterations of training a scene representation from scratch.

**These comparisons do not rest on commensurable grounds.**
On the one hand, the author utilize a model that has been subject to an extremely costly pre-training (see above), to infer the parameters of a scene representations, which are then post-processed.
On the other hand, the author utilize NeRF-based scene representations that are being trained from scratch.
In their comparison, the authors compare the time of the "post-processing" (i.e. the time to train X-Planes starting from the inferred parameters), with the time it takes to train NeRF-based scene representations from scratch.
This discrepancy is further highlighted in the comparisons on few training iterations, where X-Planes is starting from parameters that are inferred from an LRM.

X-Planes improves upon the quality of representations inferred by feed-forward models. It should hence be primarily compared (in terms of quality and time) with inferring feed-forward models.
While I do understand the reasoning behind comparing with other representations (to prove that post-training an inferred representations leads to higher quality representation than training one from scratch), the experiments should answer the question: "What is the added value of the training pipeline **compared to the outputs of the feedforward model**?".

My advice to the authors:
- Consider X-Planes as what it actually is, a post-processing method for feed-forward models. In such a case, experiments should be conducted on multiple feed-forward models, with comparisons with and without X-Planes for each model, in order to see the improvement X-Planes brings upon the output of feed-forward models.
    - In this case, the "cost" of X-Planes would be added to the inference cost of feed-forward models as this is where it would apply. While X-Planes would add significant inference costs, it could be argued that such a cost could be justified in cases where rendering quality is a priority.
- If the authors insist on considering X-Planes as a method lying in the line of work of NeRF-based scene representations, then the cost of pre-training the LRM, or any model providing the initialization of X-Planes, should be either taken into account or well justified when calculating training times, as it is a requirement for the method. In such a case, it could be argued that such a pipeline could be interesting for achieving higher quality results. However, the authors do not mention this preliminary cost.

## (w.2) The adopted evaluation protocol is ambiguous and not conclusive.

Regardless of (w.1), the evaluation protocol adopted by the authors is ambiguous and not conclusive.

- (w.2.1) The evaluation protocol is ambiguous. In section 4, the authors say that the D-NeRF dataset is not directly compatible with their method as the LRM requires a monocular video with a fixed viewpoint. Hence, they train K-Planes on the D-NeRF dataset in order to render the same viewpoint for every frame.
    - (w.2.1.q1) In this pre-processing, the "ground truth" views become the render of K-Planes. Are the authors computing their evaluation metrics against K-Planes renders?
    - (w.2.1.q2) What views is X-Planes trained and evaluated on?
    - (w.2.1.q3) In Table 1, what views are the K-Planes, Tensor4D, and other methods trained on? what views are they evaluated on?
    - (w.2.1.q4) Is the NVS setting the same between X-Planes and the other methods? (Same training/input views, same evaluation views)?
- (w.2.2) The experiments done on few-shot modeling are not directly comparable. K-Planes is not a few-view reconstruction method. The initial parameters of X-Planes are inferred from a model that does single-view to 3D reconstruction. X-Planes should be compared instead with the output of the LRM itself (in terms of quality and inference time), and potentially other feed-forward models for few-vew 3D reconstruction.
- (w.2.3) In table 1, 4D-GS is both faster and of higher quality than X-Planes.
    - (w.2.3.q1) What advantage does X-Planes have over 4D-GS?
- (w.2.4) In lines 340-341, the authors refer to an ablation study in the experiments section, which is not present. While some ablations are presented in the appendix, the ablation referred to in these lines is missing.
    - (w.2.4.q1) Could the authors provide this ablation study?
    - (w.2.4.q2) Additionally, what is the true value of the feed forward model on X-Planes? What would happen if a random initialization is given to X-Planes? An ablation of the feed-forward model seems necessary to justify the design choices.

## (w.3) The method section is not clear
- I found that the presentation of the method is not clear (especially around sections 3.2 and 3.3).
    - (w.3.q1) After the rendered K-Planes view are given to the LRM to provide the Tri-Planes, what data is used to supervise X-Planes ?
    - (w.3.q2) In equation (17), could the authors provide the entire loss X-Planes is trained with?

## (w.4) Other minor concerns
- (w.4.1) In lines 147-150, the phrase "3DGS is faster than NeRF under sufficient input views, but struggles with limited input views" is misleading. It leads to believe that NeRF does not struggle with insufficient input views, which is not the case.
- (w.4.2) In figure 2, the LRM, which is an integral part of the method, is not represented in the figure. It should be added for clarity.


## Minor typos
- In tables 1 and 2, "SSIM" should be followed by a \\uparrow ($\uparrow$).

**Questions:**

(q.1) There are no claims regarding the code base of this work. Do the authors plan on making the code for this paper open-source?

See weaknesses for the other questions.

---

### Official Review · Reviewer_kht9 · 2025-11-01

**Soundness:** 1
**Presentation:** 1
**Contribution:** 2
**Rating:** 2
**Confidence:** 3

**Summary:**

This paper aims at accelerating and enhancing the learning of dynamic 3D scenes planar representations like $K$-Planes and Tensor4D. To this end, it proposes to initialize the learned planes by supervision with the output of an image-to-triplanes pre-trained model (LRM) applied to each frame of a fixed-viewpoint video, after decomposing the concatenation of these triplanes into a low-rank (static) and sparse (dynamic) decomposition. Conducted experiments on the standard D-NeRF dataset show a better reconstructive performance and an improved efficiency for the resulting $X$-Planes compared to planar baselines.

**Strengths:**

1. The idea of leveraging a pre-trained 3D generative model to accelerate and improve NeRF training is interesting and, to my knowledge, novel. As hinted by the paper title, leveraging large pre-trained models has been crucial for modern ML advances, but this idea was not particularly explored for NeRFs, and in particular for dynamic NeRFs.
2. The proposed implementation consisting in building a better initialization for NeRF training than the default random one using triplanes inferred by the generative model is reasonable and well motivated.
3. Experimental results are promising. The method significantly improves the reconstructive performance of standard planar methods such as $K$-Planes and Tensor4D for dynamic scene modeling, while being potentially more efficient.

**Weaknesses:**

### Hidden restrictive assumptions and overclaiming

The presented method relies on restrictive assumptions that are seldom discussed in the paper, which makes it subject to overclaiming and may limit its applicability.

1. As explained in the "Dataset" paragraph of Section 4, $X$-Planes requires a monocular video with fixed viewpoint as input to the LRM. This is very restrictive and necessitates the training of $K$-Planes to initialize $X$-Planes, defeating its efficiency advantage.
2. The abstract claims that $X$-Planes "is broadly applicable to dynamic NeRF methods". This is incorrect since it can only operate with plane-based methods.
3. The decomposition of the initial planes into low-rank and sparse components relies on the strong assumption that a large part of the scene is static. Yet, the paper does not discuss the realism of this hypothesis. This raises doubts on the applicability of $X$-Planes to real-world scenes (cf. experimental weaknesses).

### Lacking experiments

The presented experiments fail to support several of the paper's main claims.

4. Other works like $K$-Planes test their method on more challenging real-world benchmarks (DyNeRF). This is not the case of this paper. Including real-world scenes would alleviate the above concerns on the method's applicability.
5. There is no ablation study justifying design choices and directly confirming that the LRM is the main source of improvement of $X$-Planes. Mainly, I would suggest the authors to replace the LRM with a $K$-Planes initialization and to train $K$-Planes and/or Tensor4D directly with the decomposition of Eq. (15) (in separate experiments).
6. While $X$-Planes outperforms other planar representations, it still cannot reach the performance of Gaussian splatting methods. The only advantage is in the few-shot setting, which represents only an extra experiment with no dedicated discussion of the related work. Hence, the experimental value of $X$-Planes compared to the SOTA is unclear.
7. The average rank experiment mentioned in lines 340-341 is nowhere to be found.
8. To facilitate visual assessment, the paper would benefit from providing animated visualizations in the supplementary material or in a dedicated anonymous website.

### Clarity and reproducibility

9. The method's description suffers from important clarity issues:
   - the optimization or initialization process leading to the conversion function $g_2$ is never full described;
   - the plane decomposition of Eq. (15) is not sufficiently motivated;
   - the paper never explains the final dynamic NeRF optimization, leaving it to the reader's guess.
10. The paper's reproducibility is lacking. No code is provided, and very few experimental details are given. Together with the aforementioned lack of clarity, this makes the proposed method hardly reproducible.

### (Minor) Formatting & typos

- Many figures are PNG images when they should be formatted using vector graphics.
- Figure 8 may not be easily readable in grayscale.
- Typos:
  - "Formutation" should be "Formulation" (l. 189);
  - "SOTA methods" should be "SOTA method" (l. 374);
  - "Both methods" is used l. 375 but its unclear which methods it refers to.

**Questions:**

Given the strong weaknesses highlighted above, I recommend to reject this paper. In my opinion, the paper cannot be reasonably improved during the rebuttal as it would require a substantial revision, hence a new round of reviewing, which is out of the scope of the discussion phase. Therefore, I encourage the authors to resubmit their work at a later conference.

Still, I remain open to discussing my recommendation with the authors and other reviewers, which I may change if my concerns or misunderstandings are alleviated in a *minor* revision. My main questions, following the highlighted weaknesses, are as follows.
1. Can the authors discuss the working assumptions and motivate the broad applicability of their method?
2. Can ablations studies and real-world scenes be included in the experiments?
3. Can the authors elaborate on the advantage of their method compared to Gaussian splatting?
4. Can more explanations be included in Section 3 and the appendix to improve clarity and reproducibility?
5. Can the authors promise to release their source code?

---

### Official Review · Reviewer_BPx9 · 2025-11-01

**Soundness:** 3
**Presentation:** 2
**Contribution:** 2
**Rating:** 4
**Confidence:** 4

**Summary:**

The paper presents a method for constructing 4D neural representations by reusing Large Reconstruction Models(LRMs). For time‑varying sequences, the LRM is applied independently to each frame to obtain a stack of 3D fields, which are then consolidated into a single 4D latent via training‑free dimensionality reduction initialized from the LRM outputs. The study demonstrates that this LRM‑based initialization plus compression alone is sufficient to achieve competitive 4D performance, avoiding the need to train a new 4D LRM and reducing computational cost, parameter count, and the number of posed training images.

**Strengths:**

- **Retrain‑free 4D construction**: Repeated LRM inference combined with training‑free compression integrates per‑frame content while maintaining temporal coherence, producing a compact 4D latent with few parameters.
- **Low‑data robustness**: The LRM‑initialized pipeline requires minimal additional parameters and no new training, providing strong initialization in time variant novel-view synthesis and improving practical portability of 4D latents.

**Weaknesses:**

- **Empirical performance vs 4D Gaussian Splatting** : Relative to 4D GS, the method underperforms in reconstruction quality and shows no significant training‑time advantage under matched compute/data budgets. Although LRM‑based initialization provides favorable starting points, the benefits are highly dependent on the multi‑plane approaches and do not translate into consistent quality or speed improvements than GS.

**Questions:**

- **Limited generality tests**: The method’s reduction should be validated beyond the presented pipeline. For instance, post‑training compression that shrinks existing multi‑plane fields (such as Tensor4D’s 9 planes → K‑Planes’ 6) without LRMs or additional neural‑rendering supervision for broader applicability.

- **Reproducibility gaps** : The dimensionality‑reduction modules lack sufficient implementation detail. Releasing full specifications and code is necessary for credible replication and downstream reuse.

---

### Note · Authors · 2025-11-20

I have read and agree with the venue's withdrawal policy on behalf of myself and my co-authors.